# Effect of Wheat Gluten and Peanut Protein Ratio on the Moisture Distribution and Textural Quality of High-Moisture Extruded Meat Analogs from an Extruder Response Perspective

**DOI:** 10.3390/foods12081696

**Published:** 2023-04-19

**Authors:** Ruixin Zhang, Yueyue Yang, Qing Liu, Liangyun Xu, Huiyi Bao, Xiaoru Ren, Zhengyu Jin, Aiquan Jiao

**Affiliations:** 1State Key Laboratory of Food Science and Technology, Jiangnan University, Wuxi 214122, China; 13526522997@163.com (R.Z.); yueyueyang@jiangnan.edu.cn (Y.Y.); liuqing9301@163.com (Q.L.); yun18356359495@163.com (L.X.); huizi_w0921@163.com (H.B.); 19851075603@163.com (X.R.); fpcenter@jiangnan.edu.cn (Z.J.); 2School of Food Science and Technology, Jiangnan University, Wuxi 214122, China; 3Collaborative Innovation Center of Food Safety and Quality Control in Jiangsu Province, Jiangnan University, Wuxi 214122, China

**Keywords:** plant-based meat, vegetable protein, high-moisture extrusion cooking, texture properties, water-holding activity

## Abstract

Wheat gluten (WG) and peanut protein powder (PPP) mixtures were extruded at high moisture to investigate the potential application of this mixture in meat analog production. Multiple factors, including the water absorption index (WAI), water solubility index (WSI), rheological properties of the mixed raw materials, die pressure, torque and specific mechanical energy (SME) during high moisture extrusion, texture properties, color, water distribution, and water activity of extrudates were analyzed to determine the relationships among the raw material characteristics, extruder response parameters, and extrudate quality. At a WG ratio of 50%, the extrudates have the lowest hardness (2.76 kg), the highest springiness (0.95), and a fibrous degree of up to 1.75. The addition of WG caused a significant rightward shift in the relaxation time of hydrogen protons in the extrudates, representing increased water mobility and water activity. A ratio of 50:50 gave the smallest total color difference (Δ*E*) (about 18.12). When the added amount of WG was 50% or less, it improved the lightness and reduced the Δ*E* compared to >50% WG. Therefore, clarifying the relationship among raw material characteristics, extruder response parameters, and extruded product quality is helpful in the systematic understanding and regulation of the fiber textural process of binary protein meat analogs.

## 1. Introduction

By 2050, the global population is expected to reach 9 billion, and the demand for meat will increase by 75% compared to today [1]. In addition, inefficient conversion causes animal meat production to occupy 83% of arable land and consume large amounts of water [2]. As a result, global environmental and food security pressures are increasing. In particular, meat consumption is faced with severe challenges, and animal meat resources are increasingly unable to fully meet the meat demand of humans [3]. Moreover, the increasing desire to address the health and ethical issues associated with the consumption of animal meat has led to a shift in dietary behavior toward vegetarianism [4]. One potential solution, high-moisture texturized vegetable protein (HMTVP), has garnered attention due to its high production efficiency and low production costs [5]. HMTVP is a meat analog that has a structure similar to meat fibers due to a four-stage formation process involving unfolding, association, aggregation, and crosslinking of vegetable protein molecular chains during the extrusion process under conditions of high moisture, high temperature, and high shear [6]. Hitherto, most studies have aimed to improve the fiber structure of extrudates by optimizing the formula and process parameters [7,8,9,10,11,12,13]. By contrast, few studies have examined the water distribution, despite the importance of water distribution for juiciness—a key characteristic of meat products [14,15,16].

Differences in raw material characteristics can also lead to great fluctuations in the extruder response parameters during the extrusion process, thereby affecting the extrudate quality. A number of previous studies have described the relationship between process parameters, extruder response, and product characteristics during high moisture extrusion [17,18]. However, for a complex system composed of multiple proteins, the relationship between raw material characteristics, extruder response parameters, and product characteristics needs to be further clarified. Therefore, it is still difficult to systematically regulate the HMTVP product by adjusting the raw materials’ characteristics and extruder response in a targeted manner.

Currently, the most common protein sources for HMTVP products are soy protein (soy isolate protein and soy concentrate protein) [19,20,21], wheat gluten [22,23], and pea protein (pea isolate protein) [24,25]. In order to improve the utilization of defatted peanut meal, peanut protein has been used in recent years. Of these, peanut protein is allergenic, with a prevalence of about 2% [26]. However, peanut protein has an excellent amino acid profile, a desirable volatile profile, a low level of antinutritional factors, and a steady supply [27]. PPP is the product of defatted peanut meal, a by-product of peanut oil production [28]. The protein, mainly composed of arachin and conarachin, contains 8 essential amino acids necessary for the human diet and has a digestibility of up to 90%. However, the fiber structure of peanut protein still needs to be further improved. Many studies have reported on the fiber textural mechanism and fiber structure improvement of peanut-protein-based meat analogs [29,30]. The fibrous degree of PPP can be promoted by the addition of polysaccharides and transglutaminase, which induce conformational changes in the protein [31,32]. 

Wheat gluten (WG) is the primary storage protein group in wheat grains [33,34]. It forms a cohesive viscoelastic network that is important in producing meat analog products. It is mainly composed of glutenin, which forms polymer networks, and gliadin, which acts as a plasticizer to improve the viscosity and ductility of wheat doughs [35]. WG can be extruded individually to form striking textures and is widely used to compound with other proteins to improve their fibrous structure. This is because it causes the mixture to form incompatibly independent phases, which are essential for the development of fibrous structures [22,36,37]. From a protein conformation perspective, increasing the WG ratio increases the content of disulfide bonds and β-sheets, resulting in a dense and structurally strong gluten network [10,23,38]. On the other hand, a strong fibrous structure can also result in a decrease in resemblance to real meat. In addition, juiciness has been neglected by many studies on the high water compression of wheat protein [39]. In other words, the extrudates obtained by high-moisture extrusion of PPP or WG alone have shortcomings in simulating the fibrous structure and juiciness of real meat.

The aim of the present study was to explore the influence of mixing WG and PPP at different ratios on extrudate quality characteristics (texture, color, and moisture distribution) from the aspect of the extruder response. The goal of this work was to explain the relationships among raw material properties, extruder response parameters, and extrudate quality to provide a basis for systematic regulation of the protein fiber textural process. Our findings support the use of binary proteins to enhance the fibrous structure of meat analogs. 

## 2. Materials and Methods

### 2.1. Materials

Peanut protein powder (PPP) and wheat gluten (WG) were supplied by Ouguo Co. (Yantai, China) and Feitian Co. (Hebi, China), respectively. The protein, starch, crude fiber, lipid, and water contents of the PPP employed in this study were 55.3% (dry basis), 5.2% (dry basis), 0.3% (dry basis), 8.8% (dry basis), and 6.1%, respectively. The protein, starch, crude fiber, lipid, and water contents of the WG employed in this study were 83.2% (dry basis), 5.7% (dry basis), 3.5% (dry basis), 1.0% (dry basis), and 8.96%, respectively. Deionized (DI) water was used throughout the experiment.

A three-dimensional movement mixer (SYH-15L, Shaoping, Wuxi, China) was used to mix the blends of PPP and WG at different ratios (dry basis). The tested WG:PPP ratios were 0:100, 30:70, 50:50, 70:30, and 100:0, respectively. A total of 12 kg of the mixture was prepared under each ratio. The conditioned mixtures were packed in plastic bags and kept at 4 °C overnight to equilibrate.

### 2.2. WAI and WSI

The water absorption index (WAI) and water solubility index (WSI) were determined using the method of Ding et al. [40] and Hirunyophat et al. [41], respectively, with appropriate modifications. A 2 g powder sample (m_0_) was weighed and placed in a dry centrifuge tube (m_1_). A 25 mL volume of distilled water was pipetted into the centrifuge tube, the tube was vortexed for 2 min and then placed in an oscillating water bath at 30 °C for 30 min, followed by centrifugation at 3500 r/min for 10 min. The tube was carefully removed from the centrifuge and placed vertically, and the supernatant was removed with a pipette gun. The supernatant was transferred to a weighing bottle (m_3_) that had been dried to a constant weight, placed in a 105 °C oven to dry to a constant weight, and then weighed (m_4_). The centrifuge tube and precipitate mass (m_2_) were weighed at the same time. Each set of samples was measured three times and averaged.

The WAI and WSI were calculated using Equation (1) and Equation (2), respectively:(1)$$\mathrm{WAI}\;\left(g/g\right)=\frac{m_{2}-m_{1}}{m_{0}}$$
(2)$$\mathrm{WSI}\;\left(\%\right)=\frac{m_{4}-m_{3}}{m_{0}}\times 100\%$$

### 2.3. Rheological Properties of Raw Materials

A 3 g sample of raw material powder was accurately weighed into a small 50 mL beaker, and the water content was adjusted to 55% based on the water content measured in Section 2.1 to simulate the hydration of raw materials in the mixing zone of the extruder. The material was thoroughly mixed, the cup was sealed with a sealing film to prevent water loss, and the powder was placed in a refrigerator at 4 °C overnight to balance the water.

The rheological properties of the samples were measured using a rotary rheometer (Discovery HR-3, TA Instruments, New Castle, DE, USA) using the following operational conditions: The mixture was placed between two 40 mm diameter plates, and the gap between the two plates was adjusted to 1050 μm. The excess sample was trimmed and removed, and a thin layer of silicone oil was applied to the edge of the sample to prevent water loss. The gap between the two plates was then adjusted to 1000 μm for the test.

#### 2.3.1. Frequency Sweep

The sample was left for 60 s, and then the frequency was scanned at 25 °C, a strain degree of 0.5%, and a frequency of 0.1–10 Hz.

#### 2.3.2. Strain Sweep

The sample was left for 60 s, and then the strain sweep was scanned at 25 °C, a vibration frequency of 1 Hz, and a strain degree of 0.1–100%.

#### 2.3.3. Creep Recovery

The sample was left for 60 s, and then stress was applied at 10 Pa for 180 s. The stress was measured at 1 Hz and 25 °C at 360 s after the stress was withdrawn.

### 2.4. Extrusion Experiments and Sample Preparation

The raw materials were extruded with a co-rotating meshing twin-screw extruder (FMHE36-24, FUMACH, Changsha, China). The raw materials were fed into the extruder at a constant speed of 6 kg/h (dry basis) by a feeder and the process was controlled by the extruder operating system. Based on the moisture content of the mixture, water is added to the extruder through a pump so that the final feed moisture content is maintained at 55%. The processing parameters of the extruder are set according to the methods of Rehrah et al. [42] and Zhang et al. [32], and adjusted on the basis of a lot of pre-experiments. The screw speed was 210 rpm. From the first to the sixth zone, the temperature was maintained at 25 °C, 60 °C, 90 °C, 165 °C, 165 °C, and 120 °C. As shown in Figure 1, a long cooling die (including the connecting and cooling parts) was connected at the cylinder die mouth of the extruder, and the temperature was controlled at 70 °C.

Once the running state of the extruder stabilized, 20–30 continuous strips of extrudates per ratio were collected, cut into 20 cm lengths, immediately vacuum sealed, and frozen at −18 °C for future analysis.

### 2.5. Extruder Responses

After the extruder reached a stable running state, the real-time control software of the extruder was used to monitor the extruder response parameters online during the extrusion process. The extruder response parameters displayed on the extruder operating system were recorded every 5 min. These parameters included the torque, die pressure, die temperature, and mass flow rate (MFR). The specific mechanical energy (SME) was calculated using Equation (3) [25]:(3)$$\mathrm{SME}\;\left(\mathrm{kJ}/\mathrm{kg}\right)=\frac{2\pi \times n\times T}{\mathrm{MFR}}$$
where n is the screw speed (rpm), and T is the torque (N·m).

### 2.6. Low-Field Nuclear Magnetic Resonance

Extrudate samples were cut into 1 cm × 1 cm pieces for low-field nuclear magnetic resonance (NMR) analysis and nuclear magnetic resonance imaging (MRI).

#### 2.6.1. Relaxation Time

We used the method of Sun et al. [43], after some adjustments in the parameter settings. The block sample was placed into a cylindrical glass tube (diameter 15 mm). The sample relaxation time (T_2_) was measured using a low-field NMR instrument (MicroMR20-030V-I, Suzhou Niumag Analytical Instrument Corporation, Suzhou, China). Carr–Purcell–Meiboom–Gill (CPMG) pulse trains were used to obtain decay signals. The main parameters of the signal acquisition were as follows: NS (scan times) = 32, NECH (echo number) = 10,000, TD (sampling point) = 249,990, TW (wait time) = 2000 ms, P2 (180° pulse time) = 12.48 μs, P1 (90° pulse time) = 6.52 μs, TE (time echo) = 0.25 ms, and SW (sampling frequency) = 100 kHz.

#### 2.6.2. Magnetic Resonance Imaging

The block sample from Section 2.2 was placed in a cylindrical glass tube (15 mm diameter) and subjected to MRI using the same low-field NMR analyzer (MicroMR20-030V-I, Suzhou Niumag Analytical Instrument Corporation, Suzhou, China). Proton density-weighted images were obtained using spin echo sequences. The echo time (TE) was 20 ms, the repetition time (TR) was 500 ms, the field of view (FOV) was 100 mm × 100 mm, and the slice width and slice gap were 1.9 mm and 2.0 mm, respectively. The pseudo-color image was obtained by software pseudo-color processing of the proton density-weighted images [44].

### 2.7. Water Activity

The water activity of the extrudates was determined using a water activity meter (FA-ST, GBX Scientific, Romans-sur-Isere, France) according to the method of Al-Jassar et al. [45]. After cooling the extruded sample to room temperature, it was cut into uniform 0.5 cm × 0.5 cm × 0.5 cm pieces. After mixing, 2 g was transferred to a small plastic dish and placed in the water activity meter at room temperature until the indicator was stable. Each set of samples was measured three times, and the values were averaged.

### 2.8. Color

The color determination was based on the method of Lee et al. [46] and Mazumder et al. [47], respectively, with appropriate modifications. The color of the extrudates was measured with a handheld colorimeter (TES-135A, Herewith TES Electrical Electronic Crop, Taiwan, China) (CIE 2° standard observers, illuminant: white LED lamp). The sample (the extrudate sample was cut into small pieces with a side length of 20 mm and a thickness of 3 mm, the raw material was spread evenly in the test dish with a thickness of 3 mm) was placed in the test dish that accompanied the colorimeter and the test dish placed on 10 layers of white paper. The CIE *L**, *a**, and *b** values of the samples were recorded using 10 randomly chosen samples. The *L** value ranged from 5 to 100. The luminance (*L** value) ranged from total darkness (*L** = 5) to white (*L** = 100). The *a** value indicated red for a positive value and green for a negative value. The yellow *b** value was positive, and blue was the negative *b** value. The extrudates in each group were measured at least 10 times, and the values were averaged. A standard white plate (parameters of CIE LAB color space: *L** = 89.73, *a** = −0.78, and *b** = 1.88) was used to calculate the color difference between the samples and the white plate. The color difference value (Δ*E*) of the extrudate was calculated using Equation (4) [18,31,46].
(4)$$\Delta E=\sqrt{{\left(L^{*}-L_{s}^{*}\right)}^{2}+{\left(a^{*}-a_{s}^{*}\right)}^{2}+{\left(b^{*}-b_{s}^{*}\right)}^{2}}$$

### 2.9. Textural Properties

A texture analyzer (TA-XT plus, Stable Micro Systems, Godalming, UK) was used to determine the extrudate texture properties, including hardness, springiness, tensile properties, and shear properties (transverse shear force, longitudinal shear force, and fiber degree), following the methods of Fang et al. [18] and Zhang et al. [32].

#### 2.9.1. Hardness and Springiness

Square blocks with the shape shown in Figure 2a were compressed to 50% of their original thickness with a P35 probe (cylinder, ∅35 mm) at a rate of 1 mm s^−1^ for 5 s, and the hardness, springiness, and chewiness were recorded. All assays were repeated 10 times, and the recorded values were averaged.

#### 2.9.2. Tensile Properties

Samples with the shape shown in Figure 2b were pulled with an A/TG probe (rigs and clamps) at a speed of 0.5 mm s^−1^. When the sample strip broke, the tensile force was recorded. All assays were repeated 10 times, and the recorded values were averaged.

#### 2.9.3. Shear Properties

Samples with the shape shown in Figure 2c were cut with an A/CKB probe (knife blade) to 75% of the original thickness at a speed of 1 mm s^−1^. The force of the cut perpendicular to the extrusion direction was the crosswise shear force (FV), and the force of the cut parallel to the extrusion direction was the lengthwise shear force (FL). The fibrous degree was expressed as the ratio of FV to FL. All assays were repeated 10 times, and the recorded values were averaged.

### 2.10. Statistical Analysis

Statistical product and service solutions (SPSS) software (Version 24.0, SPSS Inc., Chicago, IL, USA) and Origin 2022 software (Origin-Lab, Inc., Northampton, MA, USA) were used to analyze the data. Analysis of variance (ANOVA) was used to make comparisons of means. Post hoc multiple comparisons were determined by Duncan’s test, and the significance level was set at *p* < 0.05. All results are expressed as mean ± standard deviation.

## 3. Results and Discussion

### 3.1. WAI and WSI of Raw Materials with Different WG:PPP Ratios

The WAI and WSI of the raw materials at different mixing ratios are shown in Table 1. An increase in the WG ratio resulted in an increasing trend for the WAI of the raw materials, which reached a maximum of 2.55 g/g at 100% WG. This is because WG has a stronger water absorption capacity than PPP and easily forms a gluten network structure that can trap more water [48]. However, the WAI of the raw material was slightly reduced at 30% and 50% of the WG addition. This may be due to the tendency of WG to agglomerate and cross-link to form gluten network structures when mixed with water, but at higher ratios of PPP, the formation of gluten network structures is inhibited, thus preventing the gluten network from capturing more water molecules [49]. After the addition of WG exceeds 50%, when WG plays a major role in the mixture, WAI continues to rise. Therefore, increasing the WG ratio can significantly improve the WAI of the raw material. WAI also has an important effect on the rheological properties of the raw material–water mixture, causing a difference in the hydration state of the raw material. The different hydration states will affect the rheological property of the raw material and the flow state in the extruder chamber, which is very important for the interpretation of the extruder response. In addition, the strong water absorption capacity of wheat gluten may also improve the water-holding capacity of the extrudates and reduce the water dissipation of the extrudates at the exit and during storage, resulting in a more uniform water distribution.

An increase in the proportion of WG gradually decreased the WSI of the raw material, in part because of the higher solubility of PPP and because the protein content of PPP powder is lower than that of gluten powder and contains more soluble substances.

### 3.2. Rheological Properties of Raw Materials with Different WG:PPP Ratios

#### 3.2.1. Frequency Sweep Analysis

Frequency sweep analysis allows the study of the changes in the sample storage modulus G′, loss modulus G″, and loss angle tanδ with the oscillation frequency under the action of fixed stress. It is a measurement method that does not damage the structure of the sample. The frequency scanning curve of the sample at 25 °C is shown in Figure 3. Figure 3b,c show that an increase in the WG proportion caused a gradual increase in G” and a gradual decrease in tanδ to less than 1, indicating that the raw material–water mixture is more inclined to show viscous liquid characteristics (tanδ > 1) at WG proportions ≤ 70%, but shows elastic solid characteristics (tanδ < 1) when the WG proportion is 100%. This may be because WG gradually replaces PPP to become a continuous phase from A0100 to E1000, and WG enhances the three-dimensional network structure of proteins [50]. However, peanut protein is mainly composed of arachin and conarachin, which have molecular chains that expand with difficulty under conventional treatment and tend to form a viscous liquid when mixed with water [31].

#### 3.2.2. Strain Sweep Analysis

Strain sweep can reveal the linear viscoelastic region of the sample, and the transition of the linear viscoelastic region indicates the failure of the network structure. Therefore, the range of the linear viscoelastic region can be used to measure the breakdown of the three-dimensional network structure of the protein [51]. The strain scanning curves of the samples are shown in Figure 4. Figure 4b shows that an increase in the PPP proportion gradually increases the G” value, indicating the macroscopic characteristics of a viscous liquid. Figure 4c and Table 2 show that an increase in WG content gradually increases the range of the linear viscoelastic region of the raw material–water mixture. As the percentage of WG in the raw materials increased from 0% to 100%, the endpoint of the linear viscoelastic region of the raw material–water blend system increased from 0.636% to 32.019% (Table 2), indicating that WG increased the structural strength of the protein network, making it less susceptible to breakage, which is consistent with the results of Xu et al. [52]. This phenomenon will further affect the flow behavior of the melt in the extruder cavity. The larger three-dimensional structural strength of proteins may increase the degree of friction between the melt and screw and the extruder cavity wall, thereby increasing the torque and SME during extrusion.

#### 3.2.3. Creep-Recovery Analysis

The creep-recovery experiment consists of two steps. In the first step, a known stress (10 Pa) is applied to the raw material–water mixture. The mixture deforms, and the shape variable increases gradually with the passage of time. The second step is to withdraw the stress and observe the change in deformation over time, which is called the recovery stage [53]. Figure 5 shows that an increase in WG content causes a gradual increase in the creep deformation of the raw material–water mixture.

Table 3 shows the creep deformation and non-recoverable deformation of the raw material–water mixtures. When the proportion of WG reaches 100%, the creep deformation of the mixture can reach 192%, and the elastic solid features are more obvious as stronger viscoelasticity. Wu et al. [54] reported increases in the creep deformation and non-recoverable deformation of the mixture with increasing WG content, in agreement with our experimental results. This may be due to the good elongation and viscosity of the gliadin in WG and the good elasticity of the glutenin [55]. The addition of WG makes the raw material–water blend system more inclined to elastic solid characteristics, which can also be verified from the results of frequency sweep and strain sweep.

Creep deformation can therefore be used to characterize the rigidity of the protein network after stress removal. Increasing proportions of WG cause gradual increases in the non-recoverable deformation of the mixture and in the rigidity of the three-dimensional protein network. In the case of high WG ratios, the mixture has a large creep deformation and non-recoverable deformation. The melt is deformed by the screw and moves forward, but does not easily return to its original shape, which will lead to accumulation of the material in the direction of the exit and a die head pressure increase that will cause instability of the die head discharge, and this situation will have a further unfavorable impact on the quality of the extrudates.

### 3.3. Extruder Response Parameter Analysis of Raw Materials with Different WG:PPP Ratios during the High Water Extrusion Process

The extruder response parameters are an important index for monitoring the fiber textural process of a protein [29]. The extruder response can play a bridging role and establish a correlation between raw material characteristics and extrudate quality. The extruder response parameters include torque, die pressure, die temperature, and SME. In the stable operation stage of the extruder, online software was used to conduct real-time monitoring of the extruder response parameters, and the results are shown in Table 4. As the proportion of WG in the raw material system increased, the torque increased from 110.42 Nm to 116.0 Nm. Under constant conditions of screw speed, feed speed, and extruded water content, the SME is a function of torque. As shown in Table 4, the SME also increased from 699.35 kJ/kg to 759.22 kJ/kg.

Glutenin plays an important role in enhancing the structure of protein networks. With a gradual increase in the proportion of glutenin in the raw material system, the characteristics of the melt formed by the raw material and water approach those of elastic solids, so the torque and SME increase gradually. Zhang et al. [56] added a certain amount of WG with soy protein isolate (SPI) to PPP, and they found that the addition of WG increased the SME during extrusion. However, when SPI was added, the SME of the extrusion process decreased, which may be related to the difference in viscosity between SPI and WG. Zhou et al. [49] investigated the effect of soy protein on the rheological properties of wheat flour bread and found that the addition of SP diluted the gluten and reduced the viscosity of wheat dough. Some researchers have also found that SME is related to protein concentration. At higher protein concentrations, potentially denatured proteins pack tightly together and promote the entanglement and polymerization of protein molecules, which contribute to increased viscosity and thus higher SME [57].

The increase in the proportion of WG also made the material in the extruder chamber pile up in the die, so that the die pressure increased from 3.56 MPa to 4.70 MPa. This may be related to the rheological properties of the raw material. The increase in the proportion of WG causes the mixed material to become more prone to deformation and less easy to restore to its original state. The mixed material therefore accumulates at the die and causes a gradual increase in pressure at the die. The gradually increasing die pressure causes mass evaporation of extruded water at the die outlet; therefore, the total water of the extrudates gradually decreases, and some areas with serious water loss appear.

One point worth mentioning is that the system response parameters also significantly affect the texture properties and color of the extrudates. Excessive torque and SME are both important factors leading to decreased fibrous degree and darkening of color. Zhang et al. [29] found that SME was mainly controlled by the water content and the energy input mode intensity and that the fibrous degree of meat analogs could be significantly improved by changing the shear mode. Extrusion temperature is another decisive factor that determines the tensile properties and springiness, while moisture content plays an important role in the color and hardness of extrudates.

### 3.4. Low-Field NMR Analysis of Extrudates with Different WG:PPP Ratios

#### 3.4.1. Moisture Content Analysis of Extrudates with Different WG:PPP Ratios

The state and content of extruded water at different mixing ratios are shown in Table 5 and Figure 6. Wang et al. [58] found that water can be divided into three different states according to relaxation time: T_21_ (0–10 ms, strongly bound water), T_22_ (10–100 ms, weakly bound water), and T_23_ (100–1000 ms, free water). Figure 6 shows that the T_21_, T_22_, and T_23_ of the extrudates gradually increased as the proportion of WG increased to 50%, indicating that the water mobility in the extrudates increased when a small amount of WG was added, in agreement with previous research results for WG effects on water mobility in whole wheat cracker dough [59]. However, further increases in the proportion of WG resulted in a gradual decrease in water mobility. This may be because, under high die pressure, the trapped water (immobilized water) and free water in the protein network microstructure evaporate in large quantities during the discharge stage and the proportion of water with low fluidity in the total water increases.

As shown in Table 5, with the increase in WG proportion, A_21_ gradually increased and A_22_ gradually decreased, indicating that the immobilized water in the extrudates had changed to bound water. Possibly, the addition of WG promoted closer binding of the water molecules with macromolecules, thereby leading to the decrease in water mobility. Guo et al. [60] found that an increase in the content of WG in SPI-based meat analogs from 0 to 40% first caused a decrease and then an increase in T_21_ and T_22_. A deviation from that trend was evident in the present results, which may reflect differences in the response parameters of the extruder caused by the ability of the raw material itself to bind water, which affected the extruder response.

#### 3.4.2. MRI Analysis of Extrudates with Different WG:PPP Ratios

Low-field magnetic resonance imaging (MRI) was used to study the water distribution of the extrudates at different mixing ratios. Figure 7 shows the proton density images of extrudates with different mixing ratios. Brightness represents moisture content and its spatial distribution. Pseudo-color processing of the proton density image of extrudates with pseudo-color software allowed for a more intuitive depiction of the water distribution of the extrudates. Figure 8 compares the pseudo-color proton density images of extrudates with different mixing ratios, where red indicates higher water content and blue indicates lower water content.

Figure 7a–c show that the sample area has high brightness and no obvious dark areas. Figure 7d,e show that the brightness of the sample area decreases, and a large number of dark areas begin to appear. This pattern indicated that the water loss from the extrudates was not obvious at WG contents of less than 50%, but as the WG content continued to increase, the extrudates began to show significant water loss.

As shown in Figure 8a–c, the color of the sample area gradually turned red, while in Figure 8d,e, the color of the sample area gradually turned blue. The very uneven color distribution of the sample area in Figure 8e indicates that the moisture content of the sample first increased and then decreased. When the raw material system was composed entirely of WG, the water distribution was very uneven.

### 3.5. Water Activity Analysis of Extrudates with Different WG:PPP Ratios

Water activity is the ratio of the vapor pressure of water in a food product to the saturated vapor pressure of pure water at the same temperature in a confined space [61]. It represents how much water is available in the food. The water activity of the extrudates at different mixing ratios is shown in Figure 9. The addition of a small amount of WG increased the water mobility of the extrudate and improved the water activity, which was reflected in the increase in the relaxation time. Gong et al. [62] found that increasing the ratio of WG in wheat starch/gluten extruded noodles could increase the water activity of extrudates and significantly affect the water migration of extrudates, in agreement with the results of the present study. Increasing the proportion of WG beyond 50% decreased the water activity, as indicated by the decrease in the relaxation time as the WG increased from 50% to 100%. This may be related to the high die pressure and unstable discharge state.

### 3.6. Color Analysis of Extrudates with Different WG:PPP Ratios

Color is an important indicator of HMTVP product quality [63]. Too large a value for the total color difference (Δ*E*) will have a negative effect on the later deep processing and reduce the acceptance by consumers. Therefore, we investigated the correlation between the ratio of WG to PPP and the color of the extrudates.

The instrumental color measurements of the raw materials at different mixing ratios are shown in Table 6. The addition of WG reduces the *L** of the mixture and increases the *a** and *b** of the mixture, while also making the Δ*E* of the mixture larger compared to the standard white plate.

The extrudate colors at different mixing ratios are shown in Table 7. Increasing the proportion of WG from 0 to 50% significantly increased the *L** of the extrudates from 68.92 to 77.09, indicating increased brightness. This may be because an increase in the proportion of WG increases the moisture content of the extrudates. Further increases in the WG proportion decreased the extrudate *L** to 46.51, which may reflect a decrease in extrudate brightness due to water evaporation. Alternatively, it may be related to a negative effect of SME on the *L** of the extrudates [18].

Increases in the proportion of WG also significantly increased the *a** of the extrudates from 0.34 to 3.41. This may be because the addition of WG resulted in a significant increase in SME, which accelerated the Maillard reaction between amino groups and carbonyl groups [31,64]. In addition, the color of the raw material itself and its composition will also have a certain degree of influence on the color of the extrudates. Jia et al. [65] studied the effect of WG addition on the color of WG-RPC meat analogs, and found that the change in sample color was not necessarily related to the change in the dominant continuous protein phase; however, the addition of WG diluted the darker-colored RPC. In addition, oxidized tannins in RPC can form dark polymers during high-temperature processing, whereas, in the presence of WG, this oxidized tannin can react with available mercaptan groups from the WG to form colorless complexes [66]. Comparing the *b** of the raw material with that of the extrudate, it was found that extrusion caused the sample to turn more yellow in color. The increase in the proportion of WG also significantly increased the *b** of the extrudate, which may be related to the high temperatures in the extruder cavity [18].

A significant negative correlation was noted between the extrudate Δ*E* and WG content at WG proportions lower than 50%. When the proportion of WG increased from 50% to 100%, the Δ*E* of the extrudate increased significantly from 18.12 to 46.06. When the WG ratio was 50%, the extrudate was similar in color to the standard white plate.

### 3.7. Texture Properties Analysis of Extrudates with Different WG:PPP Ratios

The quality of the HMTVP was evaluated by its texture characteristics [63]. Hardness, springiness, and tensile resistance are the key factors affecting the sensorial properties of plant-based meat analogs. The fibrous degree can be used to evaluate the similarity between HMTVP and real meat. Table 8 shows the analysis results for the texture characteristics. The hardness of the WG–PPP extrudates decreased significantly from 3.51 kg to 2.76 kg, and the springiness increased from 0.92 to 0.95 when the WG proportion increased from 0 to 50%. This suggested that the increased proportion of WG gave the extrudate a soft and springy texture, in agreement with the results of Zhang et al. [39], who concluded that an increase in the WG proportion gave the SPC-WG extrudates a softer fiber. This may be because the increase in WG causes greater moisture retention, resulting in softer extrudates. Further increases in the proportion of WG significantly increased the hardness of WG-PPP extrudates to 12.81 kg and decreased the springiness to 0.91, indicating that excess WG causes the extrudate to become hard and not springy. This may be because of a positive correlation between the die pressure, SME, and WG content. As the WG is gradually increased, the discharge state of the extruder becomes unstable, and a large amount of water evaporates at the die, thereby increasing the hardness of the extrudate.

Wheat gluten (WG) is made up of glutenins and gliadins, and the gliadins can be divided into α-, γ-, and ω-gliadins according to their electrophoretic mobilities [67]. Due to their cysteine residues, the α- and γ- gliadins are linked mainly by intramolecular disulfide bonds, and form a continuous band in the presence of water during the extrusion process, giving the system viscoelasticity [68]. Therefore, the addition of excess WG resulted in a sticky extrudate lacking springiness [37].

Table 8 shows that increasing the WG ratio from 0 to 100% significantly increases the tensile strength of the WG-PPP extrudates from 0.26 kg to 0.64 kg. Zhang et al. [39] suggested that the addition of WG could improve the tensile strength of meat analogs. Similar results were observed by Jia et al. [65], who concluded that the anisotropy, tensile stress, and tensile strain of the blend were significantly improved by the addition of WG to SPC. This is probably because WG contains glutenin. Glutenin can occur as high molecular weight and low molecular weight subunits. The high molecular weight subunits form large polymers through intermolecular disulfide bonds and non-covalent interactions with gliadin, providing strength for fibrous structures [35]. Therefore, an increase in the proportion of WG leads to an increase in the strength of the protein polymeric network, resulting in a sustained increase in the tensile strength of the WG-PPP extrudates.

The crosswise shear force, lengthwise shear force, and fibrous degree of the extrudates with different mixing ratios shown in Table 8 reveal that when the WG accounted for 50% of the raw material system, the fibrous degree of the extrudates showed a positive correlation with the WG content. This may be because the glutenin in WG increases the plasticity of the extrudates during extrusion, which may result in the proteins dissolving better in the melt and forming denser structures upon cooling [23,60,69]. The inclusion of WG would also increase the disulfide bond content and facilitate cross-linking between proteins, resulting in an increased fibrous degree [10]. However, when the proportion of WG exceeded 50%, the fibrous degree of the extrudates decreased from 1.75 to 1.36. This may be because the increase in WG gradually destabilized the discharge state of the extruder, resulting in the formation of many longitudinal gaps in the extrudates. The presence of these longitudinal gaps would decrease the fibrous degree of the extrudates [39].

## 4. Conclusions

The proportions of continuous and dispersed phases in the process of high-moisture extrusion are of great significance to the mechanism that regulates the rheological properties of raw materials, the response parameters of the extruder, and the quality characteristics of the extrudates. In this study, the effects of different proportions of PPP and WG on extrudate quality characteristics were discussed from the perspective of the extruder response, and the causes of extruder response differences were discussed from the perspective of the raw material characteristics. The results support the idea that the good hydration ability of WG results in a more even distribution of water when added to the ingredients in moderation. The good moisture distribution also gives the extrudates a soft, elastic texture and white color. The introduction of glutenin also imparts a dense fiber structure and greater tensile strength to the extrudates. Conversely, when the WG ratio exceeds 50%, the good hydration ability of WG leads to the formation of dough with higher viscosity, and this prevents melt flow during extrusion, resulting in higher torque and higher SME. The large creep deformation and non-recoverable deformation of WG also lead to the accumulation of material at the die and increase the die pressure. In this case, a large amount of water evaporates at the die, the water distribution of the extrudates becomes uneven, and the extrudates become dry, hard, and dark in color. The extrudates also undergo longitudinal clearance due to the unstable discharge state, thus reducing the fibrous degree. This study found that the best quality parameters were achieved at a 50:50 balance of WG to PPP. Meanwhile, this study provides a new approach for the systematic regulation of the quality of meat analogs made from WG-PPP and other multiphase systems.

## Figures and Tables

**Figure 1 foods-12-01696-f001:**
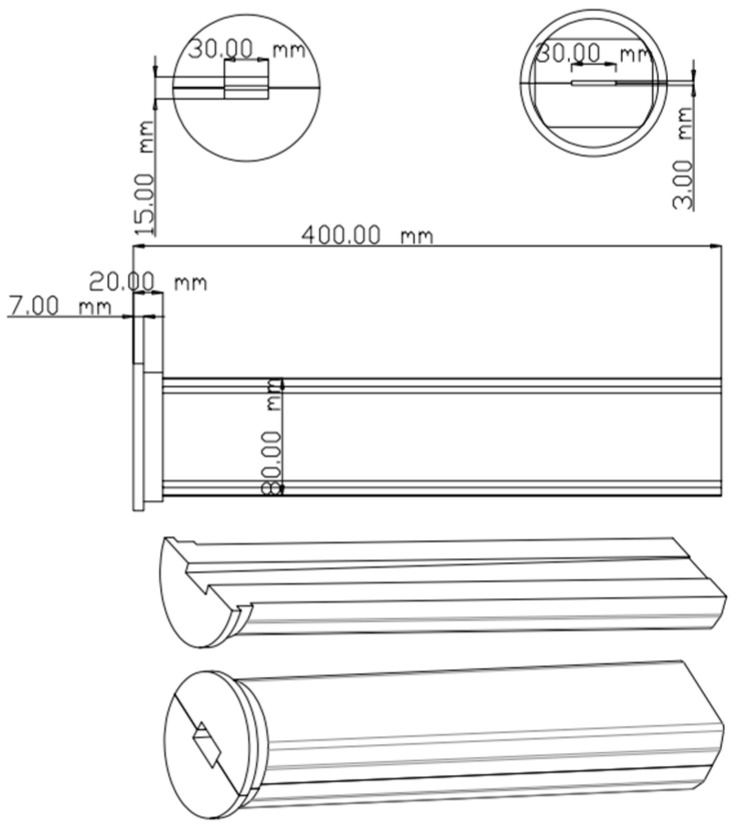
The shape and main dimensions of the long cooling die.

**Figure 2 foods-12-01696-f002:**
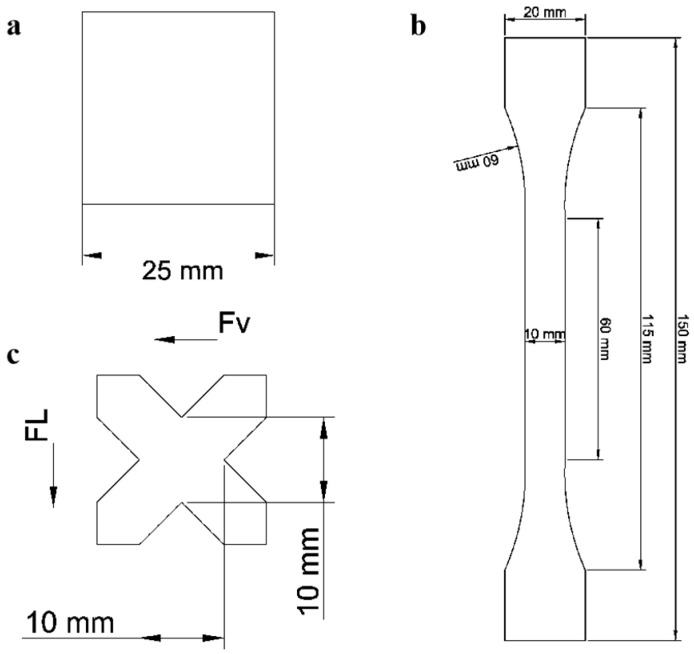
The cut shape of an extrudate used to determine hardness and springiness (**a**), tensile properties (**b**), and shear properties (**c**).

**Figure 3 foods-12-01696-f003:**
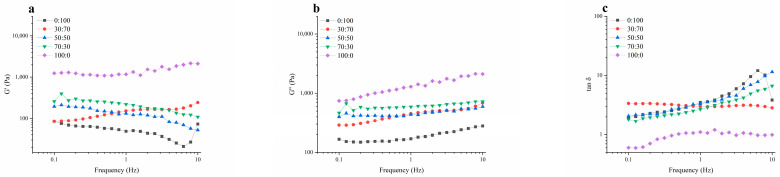
The changes in energy storage modulus G′ (**a**), energy loss modulus G″ (**b**), and tan δ (**c**) of different raw material–water mixtures under frequency sweep.

**Figure 4 foods-12-01696-f004:**
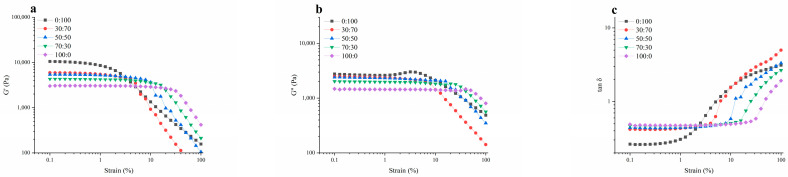
The changes in energy storage modulus G′ (**a**), energy loss modulus G″ (**b**), and tan δ (**c**) of different raw material–water mixtures under strain sweep.

**Figure 5 foods-12-01696-f005:**
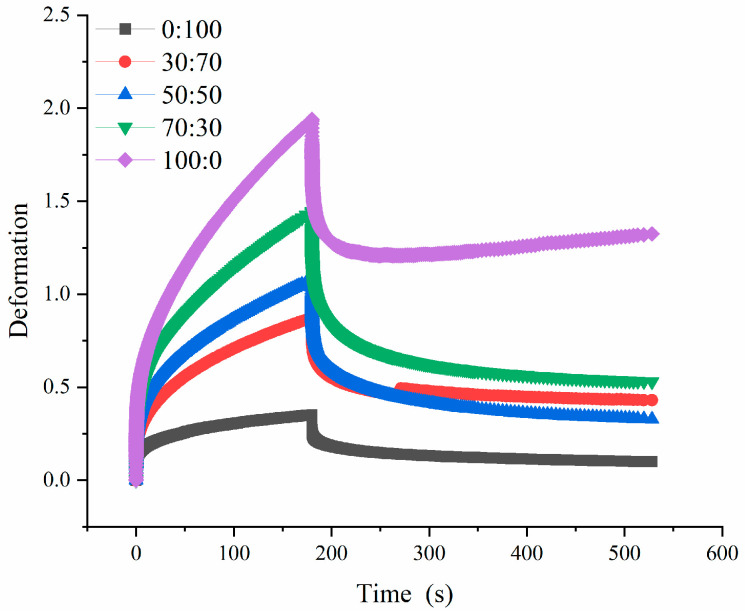
Creep-recovery curves of different raw material–water mixtures.

**Figure 6 foods-12-01696-f006:**
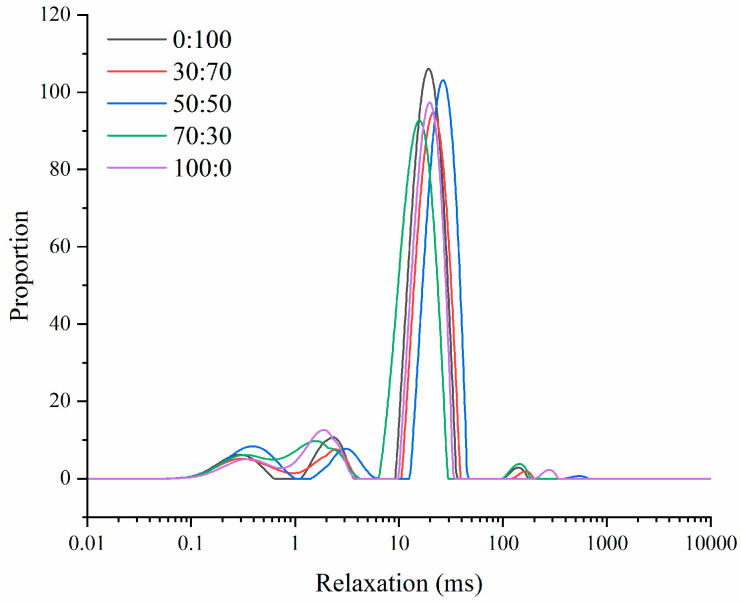
The transverse relaxation time spectra of LF-NMR of extrudates with different mixing ratios.

**Figure 7 foods-12-01696-f007:**
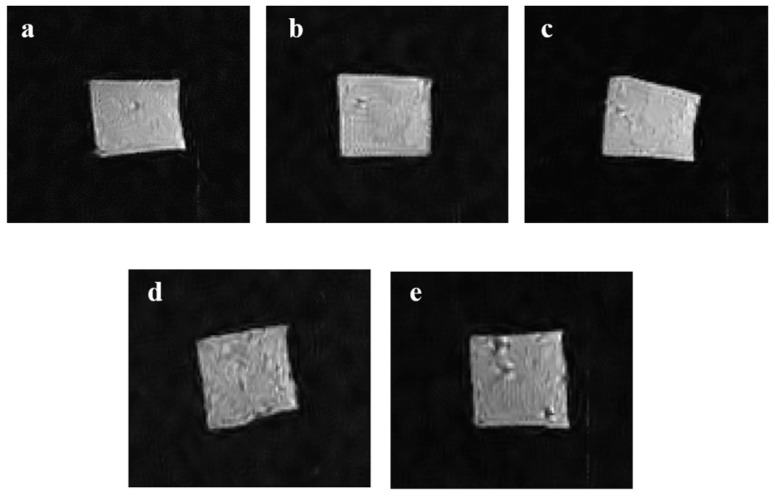
Proton density-weighted images of extrudates at the WG to PPP ratio of 0:100 (**a**); 30:70 (**b**); 50:50 (**c**); 70:30 (**d**); and 100:0 (**e**).

**Figure 8 foods-12-01696-f008:**
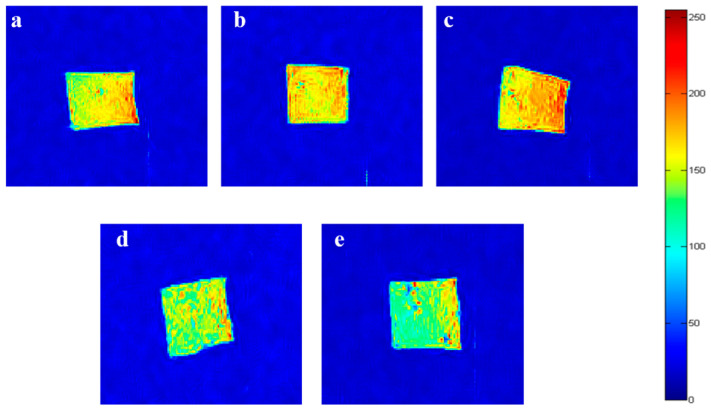
Pseudo-color proton density-weighted images of extrudates at the WG to PPP ratio of 0:100 (**a**); 30:70 (**b**); 50:50 (**c**); 70:30 (**d**); and 100:0 (**e**).

**Figure 9 foods-12-01696-f009:**
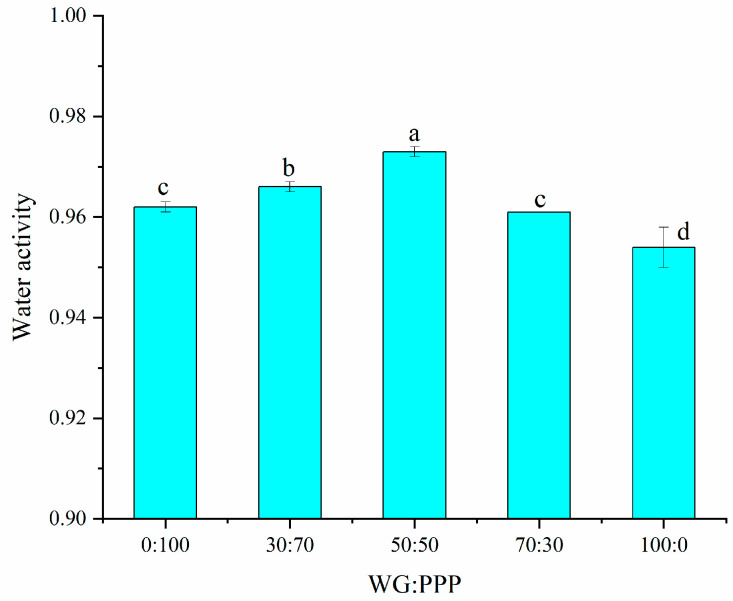
Water activity of extrudates with different mixing ratios. Different superscript letters mean significant differences (Duncan’s test; *p* < 0.05).

**Table 1 foods-12-01696-t001:** Water absorption index (WAI) and water solubility index (WSI) of different raw materials.

WG (Wheat Gluten):PPP (Peanut Protein Power) Ratio	WAI (g/g)	WSI (%)
0:100	2.10 ± 0.06 ^c^	41.64 ± 0.272 ^a^
30:70	1.88 ± 0.01 ^d^	25.12 ± 0.383 ^b^
50:50	2.04 ± 0.01 ^c^	14.77 ± 0.205 ^c^
70:30	2.24 ± 0.03 ^b^	9.06 ± 0.207 ^d^
100:0	2.55 ± 0.02 ^a^	5.83 ± 0.136 ^e^

Results are expressed as means (*n* = 3) ± standard deviation. Different superscript letters mean significant differences within the same column (Duncan’s test; *p* < 0.05).

**Table 2 foods-12-01696-t002:** Linear viscoelastic region endpoints for different ratios of raw materials.

WG:PPP Ratio	Linear Viscoelastic Region Endpoint (%)
0:100	0.636 ± 0.006 ^e^
30:70	2.011 ± 0.023 ^d^
50:50	6.373 ± 0.081 ^c^
70:30	16.250 ± 0.530 ^b^
100:0	32.019 ± 0.493 ^a^

Results are expressed as means (*n* = 3) ± standard deviation. Different superscript letters mean significant differences within the same column (Duncan’s test; *p* < 0.05).

**Table 3 foods-12-01696-t003:** Creep deformation and non-recoverable deformation of different raw materials.

WG:PPP Ratio	Creep Deformation	Non-Recoverable Deformation
0:100	0.37 ± 0.03 ^e^	0.13 ± 0.05 ^e^
30:70	0.94 ± 0.09 ^d^	0.42 ± 0.01 ^d^
50:50	1.13 ± 0.07 ^c^	0.32 ± 0.00 ^c^
70:30	1.43 ± 0.01 ^b^	0.53 ± 0.00 ^b^
100:0	1.92 ± 0.02 ^a^	1.31 ± 0.01 ^a^

Results are expressed as means (*n* = 3) ± standard deviation. Different superscript letters mean significant differences within the same column (Duncan’s test; *p* < 0.05).

**Table 4 foods-12-01696-t004:** System response parameters for extrusion processes of raw materials with different mixing ratios.

WG:PPP Ratio	Die Pressure (MPa)	Torque (N·m)	SME (kJ/kg)	Die Temperature (°C)
0:100	3.56 ± 0.11 ^d^	110.42 ± 0.38 ^e^	699.35 ± 2.39 ^e^	104.60 ± 0.55 ^d^
30:70	4.02 ± 0.08 ^c^	111.46 ± 0.36 ^d^	711.61 ± 2.28 ^d^	106.00 ± 0.71 ^d^
50:50	4.22 ± 0.04 ^b^	112.40 ± 0.63 ^c^	723.46 ± 4.07 ^c^	112.60 ± 1.14 ^c^
70:30	4.36 ± 0.05 ^b^	114.88 ± 0.18 ^b^	745.60 ± 1.16 ^b^	119.40 ± 2.88 ^b^
100:0	4.70 ± 0.20 ^a^	116.04 ± 0.71 ^a^	759.22 ± 4.66 ^a^	123.20 ± 2.59 ^a^

Results are expressed as means (*n* = 10) ± standard deviation. Different superscript letters mean significant differences within the same column (Duncan’s test; *p* < 0.05).

**Table 5 foods-12-01696-t005:** Proportions of peak areas (A_2_) of extrudates with different mixing ratios.

WG:PPP ratio	A_21_	A_22_	A_23_
0:100	11.511 ± 0.709 ^d^	87.490 ± 0.684 ^a^	0.999 ± 0.025 ^a^
30:70	12.731 ± 0.828 ^d^	86.682 ± 0.798 ^a^	0.579 ± 0.017 ^c^
50:50	14.902 ± 0.122 ^c^	84.733 ± 0.074 ^b^	0.222 ± 0.008 ^d^
70:30	19.224 ± 0.224 ^b^	79.568 ± 0.002 ^c^	0.998 ± 0.073 ^a^
100:0	21.407 ± 0.621 ^a^	77.757 ± 0.494 ^d^	0.742 ± 0.008 ^b^

Results are expressed as means (*n* = 3) ± standard deviation. Different superscript letters mean significant differences within the same column (Duncan’s test; *p* < 0.05). A_2_, the normalized peak area ratio; A_21_, the normalized peak area ratio of strongly bound water; A_22_, the normalized peak area ratio of weakly bound water; A_23_, the normalized peak area ratio of free water.

**Table 6 foods-12-01696-t006:** Color of raw materials with different mixing ratios.

WG:PPP Ratio	*L**	*a**	*b**	ΔE
0:100	89.12 ± 0.32023 ^a^	1.26 ± 0.03 ^e^	10.62 ± 0.40 ^e^	9.00 ± 0.41 ^e^
30:70	88.88 ± 0.02345 ^a^	1.48 ± 0.01 ^d^	11.88 ± 0.03 ^d^	10.29 ± 0.02 ^d^
50:50	88.548 ± 0.18754 ^b^	1.57 ± 0.02 ^c^	12.14 ± 0.12 ^c^	10.59 ± 0.10 ^c^
70:30	87.514 ± 0.21559 ^c^	1.93 ± 0.01 ^b^	13.77 ± 0.03 ^b^	12.39 ± 0.03 ^b^
100:0	86.542 ± 0.00837 ^d^	2.21 ± 0.03 ^a^	14.74 ± 0.01 ^a^	13.58 ± 0.01 ^a^

Results are expressed as means (*n* = 10) ± standard deviation. Different superscript letters mean significant differences within the same column (Duncan’s test; *p* < 0.05).

**Table 7 foods-12-01696-t007:** Color of extrudates with different mixing ratios.

WG:PPP Ratio	*L**	*a**	*b**	ΔE
0:100	68.92 ± 0.80 ^b^	0.34 ± 0.01 ^e^	13.24 ± 0.12 ^e^	23.24 ± 0.73 ^c^
30:70	70.26 ± 2.76 ^b^	0.57 ± 0.05 ^d^	13.43 ± 0.11 ^d^	22.76 ± 2.43 ^c^
50:50	77.09 ± 1.15 ^a^	0.77 ± 0.06 ^c^	14.76 ± 0.17 ^c^	18.12 ± 0.79 ^d^
70:30	51.12 ± 1.55 ^c^	2.17 ± 0.03 ^b^	15.57 ± 0.22 ^b^	41.08 ± 1.44 ^b^
100:0	46.51 ± 1.85 ^d^	3.41 ± 0.18 ^a^	17.25 ± 0.17 ^a^	46.06 ± 1.73 ^a^

Results are expressed as means (*n* = 10) ± standard deviation. Different superscript letters mean significant differences within the same column (Duncan’s test; *p* < 0.05).

**Table 8 foods-12-01696-t008:** Texture properties of extrudates with different mixing ratios.

WG:PPP Ratio	Hardness (kg)	Springiness	Tensile Resistant Force (kg)	Crosswise Shear Force (kg)	Lengthwise Shear Force (kg)	Fibrous Degree
0:100	3.51 ± 0.01 ^c^	0.92 ± 0.04 ^a,b^	0.26 ± 0.01^e^	0.50 ± 0.01 ^d^	0.43 ± 0.01 ^c^	1.18 ± 0.03 ^d^
30:70	3.48 ± 0.04 ^c^	0.93 ± 0.04 ^a,b^	0.31 ± 0.01 ^d^	0.53 ± 0.01 ^c,d^	0.36 ± 0.01 ^d^	1.47 ± 0.03 ^b^
50:50	2.76 ± 0.23 ^d^	0.95 ± 0.04 ^a^	0.37 ± 0.01 ^c^	0.56 ± 0.01 ^c^	0.32 ± 0.01^e^	1.75 ± 0.04 ^a^
70:30	6.12 ± 0.05 ^b^	0.93 ± 0.02 ^a,b^	0.41 ± 0.01 ^b^	1.19 ± 0.05 ^b^	0.81 ± 0.04 ^b^	1.48 ± 0.04 ^b^
100:0	12.81 ± 0.04 ^a^	0.91 ± 0.03 ^b^	0.64 ± 0.01 ^a^	1.61 ± 0.05 ^a^	1.19 ± 0.06 ^a^	1.36 ± 0.04 ^c^

Results are expressed as means (*n* = 10) ± standard deviation. Different superscript letters mean significant differences within the same column (Duncan’s test; *p* < 0.05).

## Data Availability

All data are contained within the article.

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
