# Peer review of "Effect of Wheat Gluten and Peanut Protein Ratio on the Moisture Distribution and Textural Quality of High-Moisture Extruded Meat Analogs from an Extruder Response Perspective"

_foods, 2023, doi:10.3390/foods12081696_

Round 1
Reviewer 1 Report
The manuscript provide good information regarding the extrusion behavior of different WG PPP blend. However, several points need to be addressed by the authors. Following the specific comments:
Line 22: indicate the parameter of the texture analysis and what the fibrous degree is.
Line 25: indicate what the “appropriate amount of WG” is.
Line 35: a reference is needed.
Line 46-47: refer to such “most studies”. Are they studying different protein source? Better expand this part
Line 56-57: this sentence is not clear to me.
Line 58-59: describe the type of raw material used and try to overview what are the most diffused ones (isolated, concentrated, dry fractionated and so on)
Lines 61-62: however, the allergenicity of such protein may be a problem for consumers… Expland this part.
Line 70: I don’t think that this can be used as a general statement. Wheat gluten is historically used for food consumption, being one of the first meat alternative in the form of seitan. Revise this part.
2.1: is the proximate analysis determined by the authors or is it from the label of the product? Specify the methods for the first case.
2.2 WAI generally is determined by an heating process. Are you sure about the determination at 30°C?
2.3: have you determined the linear viscoelastic regime before doing the frequency sweep?
Line 142: 6kg/h is not the speed but the mass flow rate. Did you determined it by a direct measurement? Specify this information in the text.
Line 201-202: what was the comparison for the delta e measurement?
Paragraph 3.1. I think the author confused the WAI with WAC. The wai is generally carried out using a heat treatments because this index measure the gelatinization behavior of the protein. WAC is the absorption of water at room temperature. Please, clarify this part.
3.2.2 The calculation of the Linear viscoelastic region must be carried out for all the mixes, reported in a table, and the discussion should be accordingly improved.
3.2.3 In this paragraph a deep discussion of the results is missing.
For the extruder response, it is not clear how the data were collected (es. 10 data recorded on what time?)
Line 340-341: this sentence is not clear. Moreover, the author should consider other studies carried out on the wheat gluten and soy rheological properties in extrusion conditions. In particular, it is well known that they differently behave in the extruder, due to a different viscosity of the material. This point need to be better clarified in the text.
To better understand the color differences, the authors should carry out the determination on the protein powders. It is still not clear to me how the delta E was calculated, in comparison with what? (Line 438-439)
May the hardness of the products be correlated with the viscosity (pressure) of the melt?
Author Response
Thank you very much for your review. Please see the attachment for the revised content and response, and thank you again for your careful review!

Reviewer 2 Report
The aim of the study was to explore the influence of mixing WG and PPP at different ratios on extrudate quality characteristics from the aspect of the extruder response. However, WG is well studied in extrusion processing, and the novel ingredient is PPP, which was also studied by Zhang et al previously. There is a lack of novelty in this study. The authors are mainly looking at how WG affect the extrudate properties, however, I think they should focus on how PPP affects the extrudate.
Table 1 - If 100% PPP has a WAI of 2.10, with the replacement of WG should increase the WAI, why at 30 and 50% replacement the WAI decreased?
It will be interesting to see the correlation on the results between the raw materials and extrudate.
Author Response

(The authors gave the same response as above.)

Reviewer 3 Report
The authors present an extensive experimental study on the effect of wheat gluten and peanut protein mixtures in extruded meat analogues, with the aim to produce a model extrudate and assess its physical characteristics.
This experimental study focus on an in-depth rheological and textural study, but the authors in the paper title use the term "sensory", in our opinion improperly, as the reader could be misled and assume that the research topic also includes various aspects of food sensory evaluation as tasting or olfactory. It would therefore be appropriate to reformulate the title by replacing "sensory quality" with "textural quality".
Paper writing and conceptualization are in line with the scope of the journal. Bibliographic sources throughout the paper appear adequate, and the topic is of interest to the scientific community and justifies the purpose of the research.
Overall scientific content of the paper shows a rigorous methodological approach, both in the choice of methodology and on testing performance. However, this reviewer will focus on his area of expertise, covering rheological texture aspects of the proposed product.
Materials and methods
Color measurement methods are adequate for product characterization; as regards the texture profile analysis methodology, it is appropriate to specify explicitly that compression test was performed as a part of a TPA test and not as a stress-relaxation test;
moreover, it is appropriate to cite, together with the reference [18] (Zhang et al., 2020) also the paper containing the original methodology (Fang et al., 2014), already reported as reference [10].
Results and Conclusions
Experimental results show significant variation between the different proportion samples, and such results may contribute to a better understanding of key parameters required for the formulation of new extrudates.
Author Response

(The authors gave the same response as above.)

Reviewer 4 Report
Foods-2311000
This paper is very well-written and with some modifications suggested below:
General comments:
How much material was used to make the analogues in total (for each of the five ratio treatments)?
How many meat analogue extrudates were made for each treatment, and what is their presentation?
How does creep and non-recoverable deformation go beyond 1 (100%)? What is the unit used as the Wu paper only reaches around 0.3?
Is there a reason why creep deformation is at 180 seconds and non-recoverable deformation is at 500 seconds?
Tables 1 to 6 – change “Samples” or “Extrudates” to “WG:PPP ratio”
Tables 1 to 6 – Footnote 2 change comma to semicolon
Tables 1 to 6 – Footnote 3 isn’t needed in the tables
Figure 3-5 – would suggest a small header for the legend
Figure 6 is difficult to interpret (as per L367-371), may be best to annotate the T21, T22, T23 on it, also should water mobility extend beyond 100%?
No mention of b* in section 3.6, this is interesting as yellow is the appropriate colour of the analogues and it increases in yellowness as more WG is added
How many extrudates from each treatment group were analysed for water activity, etc.?
In conclusion, it would be beneficial to mention that 50/50 balance was the optimal ratio for quality parameters, and why WG is a better alternative than PPP
References need either the full or abbreviated journal name, be consistent
Minor/specific comments:
L38 – change “human” to “humans”
L65 – change “in the human” to “for the human”
L88-90 – this last sentence shouldn’t be here as it’s not an aim nor background information
L117 – equation 2 should be wet powder – dry powder, therefore m3 – m4
Table 1 – is WSI a percentage, therefore 42% or 0.42%?
L145 – change “zhang [18] and [12]” to “Rehrah et al. [12] and Zhang et al. [18]”
L166 – change NMR to “nuclear magnetic resonance (MNR)”
L167 – change “NMR imaging” to “nuclear magnetic resonance imaging (MRI)”
L174-177 – space either side of =
L178 – what is SRIT? Spell out
L189 – is water activity measured as a percentage?
L193 – change “is” to “was”
L206 – What is TPA? Spell out
L208 – add “Zhang et al. » before [18]
L209 – spell out TPA
Figure 2 – suggest swapping 2b and 2c due to mention order
L229 – add “software” after 2022
L230 – how were the data analysed? ANOVA? One/two way?
L231 – change “are” to “were”
L241 – change comma to semicolon
L265 – change “less than” to “≤”
L322 – remove “, etc.”
L321-322 – add a reference for this statement
L352 – add comma after intensity
L362, 363, 395, 436 – change “WG/PPP” to “WG:PPP”
L380 – what is SPI? Spell out
L386 – add “(T2)” after time, change spectrum to spectra
L388 – define A2 in the header
L416 – add “WG to PPP” before ratio
L419– add “WG to PPP” before ratio
L437-438 – add a reference for this statement
L440 – change “WG/PPP” to “WG to PPP”
L471-473 – add a reference for this statement
L476 – change “0.91” to “0.92”
L487 – spell out WG
L511, 512, 515, 540, Table 6 – change “fibrous degree” to “degree of fibrousness”
Table 6 – change “Springiness” to “Elasticity”
Table 6 – do elasticity and degree of fibrousness have units?
L558 – add “, Tulbek, M. C. and Riaz, M. N.” after S.
L559 – at end, add”, 38.”
L568 – at end, add”, 102528.”
L570 – at end, add “, 110283.”
L572 – at end, add “, 108199.”
L587 – at end, add “, 109668.”
L592 – at end, add “, 106346.”
L595 – at end, add “, 105311.”
L605 – at end, add “, 100102.”
L609 – at end, add ”, 113561.”
L620 – at end, add “, 110099.”
L647 – at end, add “, 113546.”
L651 – at end, add “, 102758.”
References 25, 30, 43 – make LWT all capitals
Author Response
We sincerely thank you for your careful review. Please see the attached document for the amendments.

Round 2
Reviewer 1 Report
The authors managed to answer all the points expressed by the reviewers.
Author Response
Thank you for your kind words about our research!
Reviewer 2 Report
Can be accepted.
Author Response

(The authors gave the same response as above.)

Reviewer 4 Report
Foods-2311000
Paper looks much better following the revisions to the previous version. Only minor formatting comments now.
General – would suggest not starting paragraphs with an acronym
Would consider not including title words/phrases as keywords
Is Ranjbar et al. [43] still used? It appears to have been replaced by [41] and [42]
Tables 1-5, 8 – add “ratio” after WG:PPP
Specific comments:
L21 – add “a” before WG
L22 – change “elasticity” to “springiness”
L22 – change “fibrosis” to “fibrous”
L26 – change comma to full stop after parenthesis
L26 – change “is” to “was”
L27 – add “compared to > 50% WG” after ΔE
L59 – put apostrophe after materials
L63 – add “and” before pea
L66 – lower case p for peanut
L113 – add “A total of” before 12 kg
L118 – change “Hirunyopha” to “Hirunyophat”
L118 – add “, respectively,” after [42]
L154 – change “Was” to “were”
L155 – change “concrolled” to “controlled”
L165 – add “20-30” before continuous
L166 – change “were” to “per ratio were collected,”
L167 – last sentence can be removed
L187 – 2 should be subscript
L224 – change “method” to “methods” and remove second space before Fang
L246 – change “ANOVA” to “Analysis of variance (ANOVA)”
L259 – change “ratio” to “ratios”
L294 – add “energy” before loss modulus
L304 – change comma to full stop after mixture
L307 – add “(Table 2)” after 32.019%
L315 - add “energy” before loss modulus
L335 – capital t for The addition of WG…
L370 – change “SPI” to “soy protein isolate (SPI)”
L371 – capital H for However, when SPI…
L394 – change semicolon to comma after intensity
L422 – change “soybean isolate protein (SPI) -based” to “SPI-based”
L428 – change “spectrua” to spectra”
L430 – place A2 in parentheses
L484 – change “The raw materials colors” to “The instrumental color measurements of the raw materials”
L506 – add “more” before yellow, capital T for “The increase in the…”
L506-508 – add a reference for this statement
Table 6 – change “Raw materials” to “WG:PPP ratio”
Table 7 – change “Extrudates” to “WG:PPP ratio”
L527 – change “elasticity” to “springiness”
L539 – change “no springy” to “not springy”
L547-548 – change “sticky and no springy extrudate” to “sticky extrudate lacking springiness”
L617 – change “12-14” to “38”
L657 – add “, in press” after 2022
L659, 739, 741 – change “Chem” to “Chemistry”
L703, 718, 749 – change “Lwt” to “LWT”
L735 – change “Microbiol” to “Microbiology”
Author Response
Thank you for your careful review and we have made changes in the second revised manuscript in response to your suggestions. Please see the attached document for the details of response.
